# VIDEO-IN-THE-LOOP: SPAN-GROUNDED LONG VIDEO QA WITH INTERLEAVED REASONING

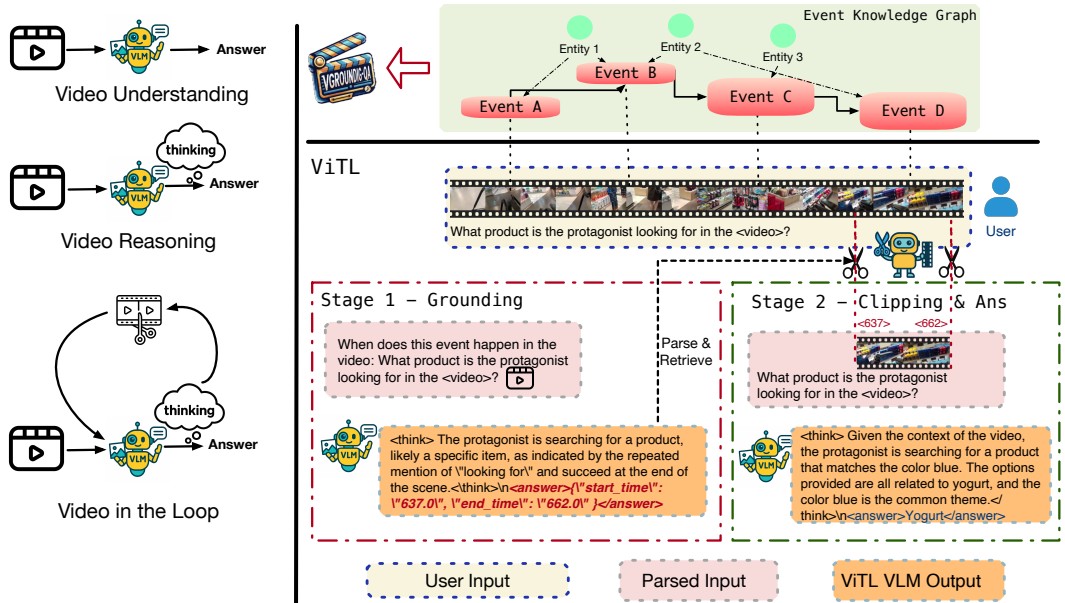

Figure 1: Overview of *ViTL* (Video-in-the-Loop) and *VGrounding-QA*. *ViTL* (right-down): Given a long video $V$ and a question $q$, **Stage 1 (Ground)** takes a *grounding query distilled from $q$* ("locate the moments needed to answer $q$") and predicts one or multiple relevant temporal spans $\mathcal{S} = \{[t_s^{(i)}, t_e^{(i)}]\}$. Supervision comes from event-graph *gold* spans. **Stage 2 (Answer)** re-encodes only frames within $\mathcal{S}$ at higher fidelity (e.g., higher frame rate/resolution) and answers the original MCQA. Training follows an R1-style loop that jointly optimizes grounding (IoU-based) and QA (cross-entropy or reward) objectives, encouraging spans that improve answering. *VGrounding-QA* (right-top): The spanning aware training set is achieved from Event Knowledge Graph.

## ABSTRACT

We present *Video-in-the-Loop* (ViTL), a two-stage long-video QA framework that preserves a fixed token budget by first *localizing* question-relevant interval(s) with a low-fps skim and then *answering* via span-aware reallocation of visual tokens at higher effective frame rate, emitting an interleaved output with both spans and the final option for direct attribution. We also introduce *VGrounding-QA*, which converts description based event graphs into *span-grounded* multiple-choice QA by pairing each question with *ground-truth* time span(s) and related reasoning. ViTL is trained end-to-end with an interleaved group-relative objective that couples temporal IoU for localization with answer correctness, allowing credit to flow from answers back to spans without increasing compute. Under fixed token budgets, ViTL attains up to 8.6% with 50% less frame input on long-video QA and temporal grounding (e.g., Charades-STA, ActivityNet-Captions) and ablations show that span-aware token reallocation consistently surpasses uniform sampling. Together, *VGrounding-QA* and ViTL provide an interpretable, compute-efficient recipe for scalable long-video QA.

# 1 INTRODUCTION

Multimodal large language models (MLLMs) have advanced rapidly (Hurst et al., 2024; Li et al., 2024a; OpenGVLab Team, 2024), showing strong performance in instruction following, open-vocabulary perception, and multi-step reasoning across images and videos. Recent systems extend temporal context windows, add memory modules, and leverage stronger backbones (e.g., 3B–70B+ vision language models), making long-form understanding an increasingly realistic goal.

Despite rapid progress, long-video QA remains brittle. Under fixed token and frame budgets, models typically adopt uniform or heuristic sampling (Team, 2025; Lin et al., 2023), spending most capacity on background and overlooking the brief moments that carry the answer. Public training sets (Gao et al., 2017; Fu et al.; Wu et al.) rarely bind each question to *ground-truth* temporal spans, so systems learn to produce answers without reliably learning *where* to look—hindering attribution and limiting evaluation beyond accuracy. Moreover, localization and answering are often optimized in isolation; even when span prediction improves (e.g., higher tIoU), those gains do not consistently translate into better QA because the learning signal does not reward spans that actually improve answers.

Recent efforts extend context windows and adopt adaptive or key-frame selection to reduce background dilution, yet many tokens still land off-target and attribution remains limited. *Temporal Video Grounding (TVG)* methods (Liu et al., b; Qu et al., 2024) strengthen localization, but their benefits rarely carry over to QA without span-aware training signals. R1–based post-training (Feng et al.; DeepSeek-AI et al.) improves step-by-step reasoning, but in the absence of span supervision and reward coupling it cannot teach *where* to search. These observations motivates us to explore a pipeline that reallocates visual tokens to evidence at fixed cost (zoom-in the video), a dataset that jointly supervises spans and answers, and a learning objective that couples localization quality with answer utility.

We present *Video-in-the-Loop* (ViTL), a two-stage procedure that allocates computation where it matters while preserving a fixed token budget. The model first performs a low–frame-rate *skim* over the entire video to localize one or more evidence intervals, then *zooms* into the predicted spans and reasons at a higher effective frame rate to produce the multiple-choice answer. Both the localized spans and the final option are emitted in a single, interleaved response, yielding direct attribution and a uniform format for evaluation. By design, ViTL turns long-video QA into a skim→zoom workflow that concentrates tokens on evidence rather than background (see Fig. 1).

We train ViTL end-to-end on the interleaved output using a group-relative policy objective (GRPO) that couples a temporal-IoU signal for span quality with an answer-correctness signal for utility. This composite reward assigns credit from the answering step back to the localization step, so spans are optimized not merely to match annotations but to improve downstream answers under the same compute. A brief supervised warm-up stabilizes decoding, after which interleaved GRPO refines both stages jointly and encourages the model to produce well-formed spans and faithful answers in one pass.

However, in long-video understanding tasks, accurately identifying the time segments relevant to a given question is highly challenging, since many questions are complex, may span multiple temporal segments, and often involve intricate temporal and object relationships. To address this, we innovatively apply *event knowledge graph* techniques: powerful MLLMs such as GPT-4o are first employed to perform fine-grained structured analysis of the entire video, extracting objects, events, and their relationships to construct an event knowledge graph. From this graph, we then select valuable nodes and edges to formulate QA questions. In this way, we can design multi-hop reasoning questions that span multiple temporal segments, which enables us to train models to simultaneously improve both temporal localization of question-relevant segments and the ability to handle complex long-video understanding problems. (see Fig. 2 )

Our contributions can be summarized as follows:

(1) We introduce a **Video-in-the-Loop** procedure that reallocates tokens to predicted evidence under a fixed budget, producing interpretable span grounding and answers.

(2) We develop a span-grounded training set construction approach and a training set called *VGrounding-QA* that ties each QA item to *ground-truth* temporal spans, supplying the missing supervision for span-aware QA.

Figure 2: **Training–set construction from event graphs via semantic chunking.** A long video is first buffered into short uniform chunks (e.g., 3s) and produces per–chunk descriptions. Neighboring chunks with high textual similarity are merged into *semantic segments*; their absolute start/end times become the **ground-truth** span(s) (example: $30:58 \rightarrow 32:19$). Each segment is summarized into an event description and converted into a *span-grounded* MCQA instance whose question is answerable using only this span; distractors are mined from other events in the same video.

(3) We perform end-to-end training that links temporal IoU and answer reward, providing direct credit assignment from answers back to spans.

(4) We evaluate on **long-video QA** and **temporal grounding**, and conduct ablations to demonstrate the effectiveness of the proposed paradigm.

## 2 RELATED WORK

### 2.1 MLLMs FOR LONG-VIDEO UNDERSTANDING

Recent multimodal LLMs extend language reasoning to video by enlarging temporal context, compressing visual tokens, and adding spatio–temporal adapters. LLaMA-VID reduces per-frame tokens to support hour-scale inputs (Li et al., 2024b); VideoLLaMA2 introduces specialized temporal connectors and an audio branch for richer dynamics (Cheng et al., 2024); and LongVA pushes sequence length into the hundred-thousand–token regime for untrimmed videos (Zhang et al., 2024b). Concurrently, video-tuned backbones (e.g., Qwen2.5-VL; Video-LLaVA) demonstrate strong zero-/few-shot results on captioning and QA (Team, 2025; Lin et al., 2023). Reasoning-centric post-training has also been explored: rule-based RL improves step-by-step solutions in text (Guo et al., 2025) and has been adapted to video settings (Feng et al., 2025). On the localization front, *temporal video grounding* integrates language with segment prediction and is commonly evaluated on Charades-STA and ActivityNet-Captions (Gao et al., 2017; Krishna et al., 2017); recent LLM-based approaches (Ren et al., 2023; Huang et al., 2024; Qian et al., 2024; Zhao et al., 2024) report stronger tIoU/Recall. While effective at broad coverage and zero-shot generalization, many systems still operate over globally sampled frames and rely on training protocols that optimize grounding and answering separately, which can leave attribution and span-to-answer transfer underexploited.

### 2.2 MULTI-STAGE AND ADAPTIVE VIDEO PROCESSING

A complementary line reduces redundancy by selecting key frames or segments before reasoning. Baselines rely on uniform or heuristic sampling, while recent methods use semantics-aware selection with VLMs such as CLIP (Radford et al., 2021)—e.g., BOLT (Liu et al., a) and AKS (Tang et al., 2025) retrieve frames most aligned with a text query and then feed them to a larger model. This improves efficiency and can outperform uniform/top-$k$ picks on long videos.

While effective, CLIP-based selection inherits a query–embedding mismatch: interrogative questions are not the distribution CLIP was trained on, so retrieved frames maximize caption-style alignment rather than evidential sufficiency for the posed question. Moreover, top-$k$ retrieval over individual frames tends to break temporal continuity—dropping onsets/offsets and interstitial motion—thereby introducing "negative" frames and missing multi-span evidence crucial for actions and causality. Because the selector is optimized with proxy salience objectives and sits upstream of answering, utility cannot flow back to refine selection, which makes performance sensitive to thresholds and limits temporal attribution.

## 3 *VGrounding-QA*: TRAINING SET CREATION FROM EVENT GRAPHS

**Overview.** We construct a span-grounded training set that couples *verifiable temporal supervision* with *answer supervision* as exemplified in Figure 2. Our pipeline proceeds in three stages: (i) selecting source annotations with broad coverage, (ii) converting each annotated event into a single MCQA instance that is answerable from its **ground-truth** temporal span(s), and (iii) performing layered quality review to ensure temporal locality and item difficulty. Table 2 shows a breif comparison with existing long video understanding datasets.

**Event Knowledge Graph construction.** We adopt event graphs built by a *semantic chunking* pipeline from prior work (Yan et al., 2025). Instead of uniform, fixed-window segmentation alone, a long video is first buffered into short, uniform chunks (e.g., ∼3s). A lightweight VLM (e.g., a 7B variant) then produces a brief description for each chunk. Neighboring chunks are compared via a text-similarity signal (BERTScore over the per–chunk descriptions); adjacent chunks with high similarity are *merged* into a single, temporally contiguous *semantic chunk*, while boundaries are enforced where similarity drops below a threshold. This yields event-level segments that better match the variable temporal granularity of real videos and remain efficient to construct under tight compute. Each semantic chunk is further summarized to obtain a concise event description, and its absolute start/end times define the event's **ground-truth** span(s). Edges between events are derived from interval relations on these spans (e.g., *Before*, *Overlaps*). Entities are retrieved and linked from each events with semantic de-duplication through embedding based clustering. Entities within the same cluster are then linked.

**Conversion to span-grounded QA.** For every event node, we first inherit and normalize its span set: timestamps are converted to seconds, spans are sorted and minimally merged when slightly overlapping, invalid ranges are removed, and disjoint occurrences are retained as multi-span. We then distill a short, entity/attribute-aware *grounding query* from the node description using GPT-o3 API (Hurst et al., 2024); deictic phrasing ("this clip/moment") is avoided so the text remains globally localizable on the timeline. Finally, we synthesize exactly one multiple-choice question (four options) conditioned on the event and its span(s). The question must be answerable using only the annotated evidence (the union when multi-span), and the three distractor options are drawn from other events in the *same video* to provide realistic, in-domain confounds. Brief rationales for span sufficiency (Stage 1) and option correctness (Stage 2) may be retained for supervised learning input.

**Quality review.** We apply a compact but strict review stack. First, schema validity ensures a single correct option and parsable fields. Second, temporal-locality checks reject items whose resolution requires frames outside the annotated span set or relies on purely holistic summaries; for multi-span events, items must genuinely require the annotated union when the narrative spans multiple segments. Third, language screening removes deictics and vague stems, keeping questions specific yet video-dependent. Fourth, a text-only screening step is applied to filter items that can be reliably answered without the video; such items are revised or discarded. Finally, near-duplicates within a video are removed, and option labels are balanced to avoid positional bias.

**Record schema.** Each instance minimally contains the fields in Table 1. Stage 1 uses the *grounding query* and *ground-truth spans* for temporal localization; Stage 2 uses the clipped segment(s) together with *question*, *options*, and *correct answer* for answer supervision.

**Splits and reporting.** Data are split *by video* to prevent leakage from shared footage. We keep domain and duration distributions comparable across splits and preserve the prevalence of multi-span items. Aggregate counts, span-length statistics (mean/median and percentiles), the proportion of multi-span instances, per-video instance counts, and option-label balance are reported in the Appendix.The resulting training set pairs **verifiable spans** for temporal grounding with **multiple-choice supervision** for answering. Same-video distractors make localization consequential under a fixed token budget, aligning the data directly with the two-stage protocol in Sec. 4.

Table 1: Minimal schema for each training instance. GT = ground truth.

| Field | Role |
|---|---|
| `video_id`, `event_id` | Link to the source video and event node |
| `time_spans` (GT) | Supervision for Stage 1 grounding (tIoU/Recall); single or multi-span |
| `event_description` | Human-readable summary of the event node |
| `grounding_query` | Span-seeking reformulation for Stage 1 localization |
| `question` | Prompt for Stage 2 reasoning on the clipped segment(s) |
| `options` (A–D), `correct_answer` | Supervision for Stage 2 answer selection |
| `stage1_reason`, `stage2_reason` | Optional signals (span sufficiency; option justification) |

Table 2: Comparison of long-video resources. "MCQA" = multiple-choice QA; "Multi-span" = multiple disjoint spans; "Reason" = per-sample reasoning fields.

| Dataset | GT spans | MCQA | Multi-span | Event-graph | Reason |
|---|---|---|---|---|---|
| Charades-STA | ✓ | | | | |
| ActivityNet-Captions (VTG splits) | ✓ | | (*rare*) | | |
| LongVideoBench / LVBench / MLVU | | ✓ | | | |
| ***VGrounding-QA* (ours)** | ✓ | ✓ | ✓ | ✓ | ✓ |

# 4 *ViTL*: INTERLEAVED TWO-STAGE GRPO WITH GROUNDED SPANS

## 4.1 TWO-STAGE VIDEO-IN-THE-LOOP WITH FRAME-LEVEL TIMESTAMP INJECTION

**Formulation.** Given a long video $V$ of duration $|V|$ and a question $Q$, the model predicts (i) a set of temporal segments $T = \{[t_s^{(m)}, t_e^{(m)}]\}_{m=1}^M$ that contain the necessary evidence, and (ii) a multiple-choice answer $A \in \{A, B, C, D\}$. Training/evaluation uses *ground-truth* spans $\mathcal{I}^*$ inherited from the event graph (Sec. 3); we allow $M \geq 1$ (multi-span).

**Stage 1: Global temporal localization.** We sample a *global* sequence of $n_g$ frames uniformly over $[0, |V|]$ to obtain $V_g = \{(x_f, t_f)\}_{f=1}^{n_g}$, where $t_f$ is the absolute time (in seconds) and $x_f$ is the image token. Conditioned on $Q$ and a short grounding query, the model outputs a structured set of segments
$$T = \{[t_s^{(m)}, t_e^{(m)}]\}_{m=1}^M, \qquad 0 \leq t_s^{(m)} < t_e^{(m)} \leq |V|,$$
together with a brief rationale. We permit $M$ to vary up to $M_{\max}$; disjoint segments are encouraged when the evidence is non-contiguous.

**Stage 2: Span-conditioned answering.** Let $U(T)$ denote the (ordered) union of predicted segments. We clip $V$ to $U(T)$ and sample a *local* sequence of $n_\ell$ frames at higher effective fps, yielding $V_\ell = \{(x_f, t_f)\}_{f=1}^{n_\ell}$ with the *same absolute timestamps* $t_f$ (no re-zeroing). Conditioned on $(Q, T, V_\ell)$ the model outputs the final option $A$ and a short justification. This reallocates visual tokens from background to evidence while keeping the total budget $n_g + n_\ell$ fixed.

**Frame-level timestamp injection (textual).** To stabilize temporal reference and enable auditable spans, each frame is serialized as an image token followed by a human-readable absolute time:
$$\langle \texttt{<image>} @ t_1 \texttt{s}, \texttt{<image>} @ t_2 \texttt{s}, \ldots, \texttt{<image>} @ t_F \texttt{s} \rangle,$$

We require Stage 1 to emit spans as ``$[\hat{t}_s, \hat{t}_e]$`` and Stage 2 to answer *using only* frames whose timestamps lie within $U(T)$. Ablations in Sec. 5.4 show that this textual timestamping improves tIoU/Recall and reduces off-by-segment errors under the same token budget.

**Budgets and sampling policy.** Unless otherwise specified, we use $n_g$ uniformly spaced frames for Stage 1 over $[0, |V|]$ and $n_\ell$ frames for Stage 2 drawn from $U(T)$ (with per-span caps to prevent degenerate allocation). Segments in $T$ are sorted and minimally merged prior to clipping; multi-span inputs are concatenated in temporal order.

**Validity constraints.** Outputs are lightly constrained at the prompt level: Stage 1 must place seconds inside `...` with numeric values (two decimals), and Stage 2 must emit exactly one option inside `<answer>...</answer>`. During training, malformed or out-of-range spans are rejected by format checks; at evaluation they are treated as invalid.

## 4.2 Learning with Interleaved Group-Relative Policy Optimization

**GRPO on interleaved sequences.** We optimize $\pi_\theta$ with Group-Relative Policy Optimization (GRPO) on interleaved outputs. For each $(Q, V)$ we sample $k$ responses $\{S_i\}_{i=1}^k$, each $S_i = [T_i; A_i]$, and compute group-relative advantages $A_i = R_i - \frac{1}{k}\sum_{j=1}^k R_j$. The objective maximizes the likelihood of higher-reward sequences:

$$\mathcal{L}_{\text{GRPO}} = -\mathbb{E}_{(Q,V)} \sum_{i=1}^k A_i \, \log \pi_\theta\big(S_i \mid V_{\text{unified}}, Q\big). \tag{1}$$

Coupling both stages within $S$ provides direct credit assignment from answer utility back to localization.

**Composite reward with span utility.** Each response $S = [T; A]$ is scored by

$$R(S) = (1 - \gamma)\, R_{\text{loc}}(T) + \gamma\, R_{\text{ans}}(A), \qquad \gamma \in [0, 1]. \tag{2}$$

Localization uses a multi-span temporal IoU against ground truth plus a small format component:

$$R_{\text{loc}}(T) = (1 - \alpha)\, \text{tIoU}\big(T, \mathcal{I}^*\big) + \alpha\, \text{Fmt}_{\text{time}}(T),$$

where tIoU is the ratio of total intersection to total union length over time; $\text{Fmt}_{\text{time}}$ rewards in-range, ordered, well-formed timestamps. The answer term rewards exact match and stable formatting:

$$R_{\text{ans}}(A) = (1 - \beta)\mathbb{1}[A = A^*] + \beta\, \text{Fmt}_{\text{ans}}(A).$$

Small $\alpha, \beta$ stabilize learning without outweighing task reward. Invalid or unparsable spans receive near-zero $\text{Fmt}_{\text{time}}$.

**Initialization, schedule, and controls.** We initialize from a base VLM (e.g., Qwen2.5-VL-3B/7B). A short supervised warm-up on MCQA *clipped to ground-truth spans* stabilizes decoding. We then run interleaved GRPO with group size $k$ (e.g., $k=3$), per-batch reward normalization, and KL control to the base policy. A light curriculum gradually increases $\gamma$ from localization-heavy to answer-balanced over early epochs. Full hyperparameters appear in Sec. 5.1.

## 5 Experiments

### 5.1 Setup

**Long-video QA benchmarks.** We evaluate on three public QA suites with multi-choice format and long, open-domain videos: **LongVideoBench** (Val split), **LVBench** (Val), and **MLVU** (Dev), where we report accuracy (%). For MLVU we additionally report the **Needle QA** subset (temporal retrieval stress test) and the macro average **M-Avg** across tasks.

**Temporal Video Grounding (TVG) benchmarks.** We use **Charades-STA** (indoor activities; trimmed queries on untrimmed videos) and **ActivityNet-Captions** (open-domain activities). Following standard practice, we report **Recall@IoU={0.3, 0.5, 0.7}** and **mIoU** (%).

**Role of our event-graph dataset.** Our event-graph grounded dataset (Sec. 3) provides *training* and *diagnostic* supervision: each question is paired with *gold* time span(s) derived from an event graph, enabling learnable temporal grounding and auditable evaluation. Unless otherwise specified, this dataset is not used as a held-out test set; all reported results use the official public splits of the benchmarks above.

**Evaluation protocol and fairness.** To ensure comparability, all systems run under matched compute budgets. We report (when applicable) *frames* per question and keep the total token/FLOPs

budget fixed across baselines. Unless stated, we do not use external subtitles/ASR. Inference for *ViTL* is two-stage: a low-fps global sweep for *localization* (Stage 1), followed by span-aware high-fidelity *answering* (Stage 2). Results are reported on 4×A100 80G; hyper-parameters are provided in the Appendix for reproducibility.

**Backbones.** We instantiate *ViTL* with **Qwen2.5-VL 3B** and **Qwen2.5-VL 7B**. Unless otherwise noted, Stage 1 and Stage 2 share the same backbone. In a compute-efficient setting, we also study a light *selector* (3B) with a stronger *answerer* (7B) in Sec. A.

**Training protocol.** We adopt a simple three-step schedule that closes the loop between localization and answering while keeping budgets explicit. **(T1) Supervised warm-up.** *Grounding SFT*: train Stage 1 with IoU-based span supervision on our event-graph dataset (gold span(s) per question); we minimize a boundary/IoU loss. *QA SFT*: train Stage 2 on the paired MCQA (cross-entropy over options), using only frames within the *gold* spans. For TVG benchmarks, we do not fine-tune on their train splits unless explicitly marked as FT. **(T2) Two-stage coupling (teacher forcing).** We connect the stages by feeding Stage 1 predictions into Stage 2. To stabilize learning, we sample a fixed ratio of teacher-forced spans (gold) and model spans (predicted) during training, and restrict Stage 2's visual budget to the union of selected span(s). **(T3) R1-style post-training.** We optimize a weighted reward that encourages spans which *improve answering*: *TVG reward* = IoU with gold span(s) (shape with thresholds), *QA reward* = 1 for correct option and 0 otherwise (plus mild format/length penalties). We apply PPO with a KL penalty to the SFT reference; gradients update both stages. The overall objective is a weighted sum of TVG and QA terms; ablations over these weights are in Sec. 5.4.

**Inputs and budgets.** Stage 1 consumes a fixed number of frames (e.g., 64 frames) and takes the global view of the full video and a *grounding query* distilled from the question ("locate the moments needed to answer $q$"). Stage 2 re-encodes only the predicted span(s) at higher fidelity (e.g., 4–8 fps and/or higher resolution), up to $K$ spans (default $K=5$), keeping the *total* token/frame budget matched to baselines. Unless noted, Stage 2 answers the original MCQA (no extra hints) on the clipped segment(s).

## 5.2 LONG-VIDEO QA PERFORMANCE

We evaluate the full *ViTL* pipeline on **LongVideoBench** (Val), **LVBench** (Val), and **MLVU** (Dev). Unless noted, comparisons follow the no-subtitles, matched-preprocessing setting (global low-fps sweep; fixed resolution) for fairness. Our improvements stem from two ingredients: (i) an *event-graph* dataset that pairs reasoning-centric questions with **ground-truth** time spans (Sec. 3), enabling learnable and auditable temporal grounding; and (ii) a *two-stage, R1-style* training protocol (Sec. 4) that couples IoU-based grounding signals with QA objectives, encouraging spans that *actually* improve answering under fixed token/frame budgets.

**Discussion.** The largest gains appear on **LVBench**, where relevant moments are sparse and uniform sampling wastes budget. By learning spans from event-graph supervision and coupling them to answering with a two-stage R1-style objective, *ViTL* reallocates visual tokens toward evidence—achieving higher accuracy at comparable (or lower) frame budgets. An oracle-span analysis in Sec. 5.4 further quantifies the remaining headroom from localization fidelity, supporting both the dataset design and the training protocol.

## 5.3 TEMPORAL GROUNDING PERFORMANCE

We then assess *ViTL*'s ability to *localize* question-relevant moments. All results are zero-shot unless noted. The gains are primarily driven by two factors: (i) our *event-graph* dataset that pairs reasoning-centric questions with *groundtruth* time spans (Sec. 3), providing auditable supervision for temporal grounding; and (ii) our *two-stage, R1-style* training that jointly optimizes an IoU-based grounding signal and a QA objective, encouraging spans that improve answering (Sec . 4.2.

**Charades-STA (Gao et al., 2017).** This benchmark focuses on indoor activities with natural language queries. As shown in Table 4, *ViTL* (**Ours 7B**) surpasses specialized VTG systems across

Table 3: **Long-video QA benchmarks**. Accuracy (%) on **LongVideoBench** (Val), **LVBench** (Val), and **MLVU** (M-Avg). "Frames" is the per-question frame budget when available. [†] official numbers; [‡] our re-test under the matched preprocessing (global low-fps sweep; uniform sampling; 448 resolution). "—" indicates not reported under the same setting.

| Models | Size | Frames | LongVideoBench | LVBench | MLVU M-Avg |
|---|---|---|---|---|---|
| *Closed Video MLLMs* | | | | | |
| GLM-4V-Plus[†] | – | 256 | 70.8 | 58.7 | – |
| GPT-4o[†] | – | 384 | 66.7 | 27.0 | 64.6 |
| Gemini-1.5-Pro[†] | – | 0.5 fps | 64.0 | 33.1 | – |
| *Small Video MLLMs* | | | | | |
| VITA-1.5 | 7B | 16 | 56.1 | – | – |
| LLaVA-Video | 7B | 64 | 58.2 | – | – |
| LongVA | 7B | 128 | 52.1 | 39.4 | 52.0 |
| NVILA | 8B | 256 | 57.7 | – | – |
| ByteVideoLLM | 14B | 256 | – | – | – |
| VideoLLaMA3 | 17B | 180 | 59.8 | 45.3 | – |
| InternVL3 | 8B | 16–64 | 62.5 | – | – |
| Qwen2.5-VL[†] | 7B | 256 | 56.0 (224 res) | 45.3 | 54.5 |
| Qwen2.5-VL[‡] | 7B | 256 | 61.8 (448 res) | 43.7 | – |
| *ViTL* (Qwen2.5-VL) | **7B** | **128** | **63.3** | **47.4** | **62.3** |

Table 4: Zero-shot temporal grounding on Charades-STA (Gao et al., 2017). **Bold** marks our scores.

| Method | Size | R@0.3 (%) | R@0.5 (%) | R@0.7 (%) | mIoU (%) |
|---|---|---|---|---|---|
| VTimeLLM (Huang et al., 2024) | 13B | 55.3 | 34.3 | 14.7 | 34.6 |
| TimeChat (Ren et al., 2023) | 7B | 51.5 | 32.2 | 13.4 | – |
| Momentor (Qian et al., 2024) | 7B | 42.6 | 26.6 | 11.6 | 28.5 |
| HawkEye (Zhao et al., 2024) | 7B | 50.6 | 31.4 | 14.5 | 33.7 |
| ChatVTG (Qu et al., 2024) | 7B | 52.7 | 33.0 | 15.9 | 34.9 |
| VideoChat-TPO (Yan et al., 2024) | 7B | 58.3 | 40.2 | 18.4 | 38.1 |
| E.T. Chat (Liu et al., 2024b) | 4B | 65.7 | 45.9 | 20.0 | 42.3 |
| *ViTL* (Ours 7B) | **7B** | **77.7** | **63.5** | **36.3** | **54.0** |

recall thresholds and mIoU. **ActivityNet-Captions (Krishna et al., 2017).** Results on open-domain, untrimmed videos are shown in Table 5. *ViTL* (**Ours 7B**) maintains strong localization across R@IoU levels.

## 5.4 ABLATION STUDIES ON LONG-VIDEO QA

We quantify which ingredients drive gains under *matched token/frame budgets* and a fixed backbone (Qwen2.5-VL 3B/7B). We report accuracy (%) on **LongVideoBench** and **LVBench** . All single-stage variants consume $n_g + n_\ell$ frames uniformly over the full timeline; the two-stage system uses $n_g$ (global skim) + $n_\ell$ (zoomed evidence).

**Settings.** A) **SFT** — single-stage supervised fine-tuning on our MCQA (no timestamps). B) **SFT + TI** — single-stage with *frame-level textual timestamp injection*. C) **SFT + Stage-2-only + TI (full video)** — no Stage 1; apply the Stage-2 answering prompt with timestamps to the *entire video* (no cropping); total frames $n_g + n_\ell$ sampled uniformly. D) *ViTL*(**full**) — two-stage with interleaved GRPO (coupled QA+TVG rewards) and timestamp injection in both stages.

**Findings.** *Timestamping improves global reasoning.* A→B isolates frame-level time tokens in a single-stage setup and yields consistent gains on both benchmarks, indicating reduced temporal ambiguity when evidence is dispersed. *Answer-centric training helps without localization.* B→C shows that the Stage-2 answering prompt with timestamps—applied to the full video at the same budget—further boosts accuracy, suggesting better temporal utilization despite no cropping. *Full ViTL is best at fixed compute.* C→D adds learned localization and coupled QA+TVG rewards via in-

Table 5: Zero-shot temporal grounding on ActivityNet-Captions (Krishna et al., 2017). FT indicates fine-tuning on the downstream training split. **Bold** marks our scores.

| Method | Size | FT | R@0.3 (%) | R@0.5 (%) | R@0.7 (%) | mIoU (%) |
|---|---|---|---|---|---|---|
| 2D-TAN (Zhang et al., 2020) | – | ✓ | 60.4 | 43.4 | 25.0 | 42.5 |
| MMN (Zhang et al., 2021) | – | ✓ | 64.5 | 48.2 | 29.4 | 46.6 |
| VDI (Luo et al., 2023a) | – | ✓ | – | 48.1 | 28.8 | – |
| VideoChat (Li et al., 2023) | 7B | ✗ | 8.8 | 3.7 | 1.5 | 7.2 |
| Video-LLaMA (Zhang et al., 2023) | 7B | ✗ | 6.9 | 2.1 | 0.8 | 6.5 |
| Video-ChatGPT (Maaz et al., 2023) | 7B | ✗ | 26.4 | 13.6 | 6.1 | 18.9 |
| Valley (Luo et al., 2023b) | 7B | ✗ | 30.6 | 13.7 | 8.1 | 21.9 |
| ChatVTG (Qu et al., 2024) | 7B | ✗ | 40.7 | 22.5 | 9.4 | 27.2 |
| Momentor (Qian et al., 2024) | 7B | ✗ | 42.9 | 23.0 | 12.4 | 29.3 |
| E.T. Chat (Liu et al., 2024b) | 4B | ✗ | 24.1 | 12.8 | 6.1 | 18.9 |
| *ViTL* (Ours 7B) | **7B** | ✗ | **55.1** | **46.3** | **30.0** | **24.1** |

Table 6: **Ablations on long-video QA (fixed token budget).** Accuracy (%). Higher is better.

| Config | LongVideoBench ↑ | LVBench ↑ |
|---|---|---|
| A) SFT (single, no TI) | 61.3 | 40.2 |
| B) SFT + Timestamp Injection (single) | 62.1 | 44.3 |
| C) SFT + Stage-2-only + TI (full video; no crop) | 62.5 | 45.4 |
| D) *ViTL*(full): two-stage + GRPO (QA+TVG) + TI | **63.5** | **47.9** |

terleaved GRPO. Accuracy improves on both datasets, confirming that allocating tokens to predicted evidence (while keeping totals fixed) yields better answers than any single-stage alternative.

## 5.5 EFFECTIVENESS OF LEARNED GROUNDING

To validate the necessity of a learned grounding module, we compare *ViTL* against heuristic baselines and establish performance bounds.

**Parametric vs. Non-Parametric Frame Selection.** We compare *ViTL* against a strong CLIP-based baseline (ViT-L/14), which computes the cosine similarity between the question and every frame, selecting the top-128 frames. As shown in Table 7, *ViTL* consistently yields higher accuracy (e.g., 63.3% vs. 58.1% on LongVideoBench), demonstrating that query-conditioned parametric selection provides essential evidence that zero-shot semantic similarity misses.

**Performance Bounds: Random vs. Oracle Zooming.** To quantify the headroom for Stage 1 localization, we establish lower bounds (random sampling) and upper bounds (oracle spans). Table 8 shows that *ViTL* significantly outperforms random zooming and closes a large portion of the gap toward the oracle upper bound (e.g., capturing over half the headroom on LongVideoBench). This confirms gains are attributable to accurate temporal localization.

Table 7: Parametric (ViTL) vs. Non-parametric (CLIP) selection.

| Method | Strategy | LVideoB | LVBench |
|---|---|---|---|
| CLIP Top-$k$ | Non-param. | 58.1 | 40.3 |
| *ViTL* | **Parametric** | **63.3** | **47.4** |

Table 8: Performance bounds analysis: Random vs. Oracle.

| Config | Input | LVideoB | LVBench |
|---|---|---|---|
| Lower Bound | Random | 55.2 | 39.5 |
| *ViTL* | **Pred. Spans** | **63.3** | **47.4** |
| Upper Bound | GT Spans | 70.5 | 51.4 |

## 5.6 MECHANISM ANALYSIS: ZOOMING AND MULTI-SPAN RETRIEVAL

We further disentangle the architectural decisions that drive *ViTL*'s performance.

**Impact of Visual Zooming vs. Iterative Reasoning.** To verify that gains stem from accessing high-fidelity visual details rather than simply "thinking twice," we introduce a *Refine-only* baseline (Stage 2 reuses Stage 1's low-fps frames). As summarized in Table 9, Refine-only provides only marginal gains (56.2% vs. 56.0%), whereas *ViTL* with true Zoom achieves substantial improvements (63.3%). This proves that reallocating the visual token budget to high-resolution evidence is the primary driver of performance.

Table 9: Mechanism ablation (Zoom vs. Refine). Iterative reasoning alone (Refine-only) yields negligible gains; improvements stem from high-fidelity visual zooming.

| Method | Mechanism | LVideoBench | LVBench |
|---|---|---|---|
| Single Stage | Direct answer | 56.0 | 45.3 |
| Refine-only | Think twice (low FPS) | 56.2 | 45.7 |
| *ViTL* | **Think twice (Zoom)** | **63.3** | **47.4** |

**Single vs. Multi-Span Retrieval.** Allowing the retrieval of multiple disjoint spans yields a **+2.3%** absolute accuracy gain on LVBench (47.4% vs. 45.1% with a single-span constraint). This indicates that relevant evidence in long-form videos is frequently scattered, validating our multi-span design.

## 5.7 QUALITATIVE ANALYSIS

To provide further insight into the behavior and capabilities of *ViTL*, Figure 1 illustrates representative an example of our model's two-stage reasoning process, showcasing its ability to generate coherent thought processes, accurately ground temporal segments, and provide correct answers. Additional qualitative examples, including comparisons with baselines and failure case analyzes, are provided in the appendix A.

## 6 CONCLUSION AND DISCUSSION

We cast long-video QA as allocating a fixed token budget to verifiable evidence and introduced *Video-in-the-Loop* (ViTL), a skim→zoom pipeline that first *localizes* evidence spans and then *answers* within them. ViTL trains an interleaved span+answer output end-to-end with a group-relative objective coupling temporal IoU and answer correctness, enabling credit to flow from answers back to localization. To supply supervision, we develop event knowledge graphs based approach to turn video into span-grounded MCQA that ties each question to *ground-truth* time span(s) with same-video distractors. Together, the data and method move tokens off background, make "where" explicit, and improve QA and grounding under matched compute. Limitations include noise in upstream graphs, the simplicity of MCQA versus open-ended reasoning, and RL variance; many real scenarios also require richer audiovisual cues. Promising directions include streaming ViTL (online skim→zoom), multi-hop reasoning across events/videos, space–time grounding with entity tracks, preference/rationale supervision for faithful attribution, and joint metrics scoring answer quality, span faithfulness, and compute.

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

## A  MORE EVALUATION RESULTS 7B MODELS

We further enhance Qwen2.5-VL-7B model with our two-stage RL training pipeline *ViTL* and compare the results on temporal grounding task and QA answering task on Charade-STA (Gao et al., 2017) and CG-Bench (Chen et al., 2024a).

**Charades-STA (Gao et al., 2017).**  As presented in Table 10, *ViTL* (**Ours 7B**) achieves strong performance, outperforming several specialized methods in mIoU and recall at various IoU thresholds. The *ViTL* (**Ours 3B**) variant also shows competitive results.

Table 10: Zero-shot Video Temporal Grounding on Charades-STA (Gao et al., 2017). **Bold** indicates our model's results and its scores.

| Method | Size | R@0.3 (%) ↑ | R@0.5 (%) ↑ | R@0.7 (%) ↑ | mIoU (%) ↑ |
|---|---|---|---|---|---|
| VTimeLLM  (Huang et al., 2024) | 13B | 55.3 | 34.3 | 14.7 | 34.6 |
| TimeChat  (Ren et al., 2023) | 7B | 51.5 | 32.2 | 13.4 | – |
| Momentor  (Qian et al., 2024) | 7B | 42.6 | 26.6 | 11.6 | 28.5 |
| HawkEye  (Zhao et al., 2024) | 7B | 50.6 | 31.4 | 14.5 | 33.7 |
| ChatVTG  (Qu et al., 2024) | 7B | 52.7 | 33.0 | 15.9 | 34.9 |
| VideoChat-TPO  (Yan et al., 2024) | 7B | 58.3 | 40.2 | 18.4 | 38.1 |
| E.T. Chat  (Liu et al., 2024b) | 4B | 65.7 | 45.9 | 20.0 | 42.3 |
| *ViTL* (**Ours 3B**) | **3B** | **77.7** | **63.5** | **36.3** | **54.0** |
| *ViTL* (**Ours 7B**) | **7B** | **80.1** | **66.0** | **40.2** | **59.0** |

**CG-Bench (Chen et al., 2024a).**  Table 11 shows that *ViTL* (**Ours 3B**) and *ViTL* (**Ours 7B**) achieves competitive mIoU for grounding compared to other models of similar and larger sizes, while also providing a strong long-form accuracy.

Table 11: Grounded VideoQA performance on CG-Bench. Closed-source APIs (top) and open-source models (bottom) are grouped separately and sorted by parameter size. **Bold** indicates our model's results and its scores. ViTL (3B/7B) consistently outperforms open-source baselines of similar and larger sizes in grounding quality (mIoU).

| Method | Size | Type | long-acc. (%) ↑ | mIoU (%) ↑ |
|---|---|---|---|---|
| *Closed-source (API)* | | | | |
| Gemini-1.5-Flash  (Team et al., 2024) | – | API | 32.3 | 3.67 |
| GPT-4o-mini  (OpenAI, 2024) | – | API | 33.4 | 3.75 |
| Gemini-1.5-Pro  (Team et al., 2024) | – | API | 37.2 | 3.95 |
| Claude-3.5-Sonnet  (Anthropic, 2024) | – | API | 40.5 | 3.99 |
| GPT-4o  (OpenAI, 2024) | – | API | 45.2 | 5.62 |
| *Open-source models* | | | | |
| Qwen2.5VL-instruct  (Team, 2025) | 3B | Open | 18.4 | 0.86 |
| *ViTL* **(Ours 3B)** | **3B** | Open | **23.5** | **2.90** |
| Video-LLaVA  (Lin et al., 2023) | 7B | Open | 16.2 | 1.13 |
| VideoLLaMA 2  (Zhang et al., 2023) | 7B | Open | 18.4 | 1.21 |
| Videochat2  (Li et al., 2023) | 7B | Open | 19.3 | 1.28 |
| Qwen-VL-Chat  (Bai et al., 2023) | 7B | Open | 21.6 | 0.89 |
| ST-LLM  (Liu et al., 2023) | 7B | Open | 23.8 | 2.23 |
| LongVA  (Zhang et al., 2024a) | 7B | Open | 28.7 | 2.94 |
| LLaVA-OV  (Li et al., 2024a) | 7B | Open | 31.1 | 1.63 |
| *ViTL* **(Ours 7B)** | **7B** | Open | **34.4** | **3.32** |
| MiniCPM-v2.6  (Yao et al., 2024) | 8B | Open | 30.1 | 2.35 |
| Kangaroo  (Liu et al., 2024a) | 8B | Open | 30.2 | 2.56 |
| Chat-UniVi-v1.5  (Jin et al., 2023) | 13B | Open | 25.9 | 2.07 |
| Video-CCAM  (Fei et al., 2024) | 14B | Open | 29.7 | 2.63 |
| ShareGPT4Video  (Chen et al., 2024b) | 16B | Open | 26.7 | 1.85 |
| VITA  (Fu et al., 2024) | 8×7B | Open | 33.3 | 3.06 |
| Qwen2-VL  (Wang et al., 2024) | 72B | Open | 41.3 | 3.58 |
| InternVL2  (OpenGVLab Team, 2024) | 78B | Open | 42.2 | 3.91 |

# B   ADDITIONAL ANALYSES AND DISCUSSIONS

## B.1   ROBUSTNESS TO TEMPORAL GROUNDING BIAS AND QVHIGHLIGHTS TRANSFER

A recent line of work shows that temporal grounding models often exploit dataset-specific priors (e.g., typical moment location and duration) instead of learning true cross-modal reasoning. To verify that *ViTL* does not rely on such priors, we evaluate it on ActivityNet-CD, an out-of-distribution (OOD) split of ActivityNet-Captions specifically constructed to break these biases, and further perform zero-shot moment retrieval on QVHighlights.

As shown in Table 12, *ViTL* (7B) maintains strong performance under distribution shift: its mIoU drops only slightly from 24.1 to 23.3 when moving from the in-domain ActivityNet-Captions split to ActivityNet-CD, while still clearly outperforming the Qwen2.5-VL-7B baseline in both settings. On QVHighlights zero-shot transfer, *ViTL* also improves mAP@5 by +1.3 absolute (14.2 vs. 12.9), supporting that our "Skim–Zoom" pipeline learns query-conditioned grounding rather than memorizing temporal priors.

Table 12: In-domain vs. distribution-shift generalization and zero-shot transfer. *ViTL* maintains strong performance on ActivityNet-CD (OOD) and improves zero-shot moment retrieval on QVHighlights.

| Dataset | Setting | Metric | Qwen2.5-VL-7B | *ViTL* (7B) |
|---|---|---|---|---|
| ActivityNet-Captions | In-domain (Train = Test dist.) | mIoU | 22.5 | **24.1** |
| ActivityNet-CD | OOD (distribution-shift) | mIoU | 20.2 | **23.3** |
| QVHighlights | Zero-shot moment retrieval | mAP@5 | 12.9 | **14.2** |

## B.2 Open-Ended QA Capabilities

While our main experiments adopt multiple-choice QA for standardized evaluation, *ViTL* is built on top of generalist MLLMs (Qwen2.5-VL) and naturally supports open-ended generation. To assess these generative capabilities, we evaluate *ViTL* on EgoTempo, an open-ended QA benchmark.

Table 13 shows that *ViTL* improves open-ended QA accuracy from 26.1% (Qwen2.5-VL-7B) to 31.0%, demonstrating that our "Skim–Zoom" temporal grounding also benefits free-form reasoning and not only multiple-choice formats.

Table 13: Open-ended QA performance on EgoTempo. *ViTL* improves open-ended QA accuracy over the Qwen2.5-VL-7B backbone.

| Model | Task | Accuracy (%) |
|---|---|---|
| Qwen2.5-VL-7B | Open-ended QA | 26.1 |
| *ViTL* (Ours) | Open-ended QA | **31.0** |

## C  More Qualitative Results

We provide additional qualitative examples of our video-in-the-loop approach in Figure 3, Figure 4, Figure 5 and Figure 6

## D  Declaration of LLM Usage

We used large language models (LLMs) solely for light editing of prose—including wording refinement, grammar correction, and minor clarity improvements—in limited portions of this paper. All LLM-edited text was subsequently reviewed and revised by the authors.

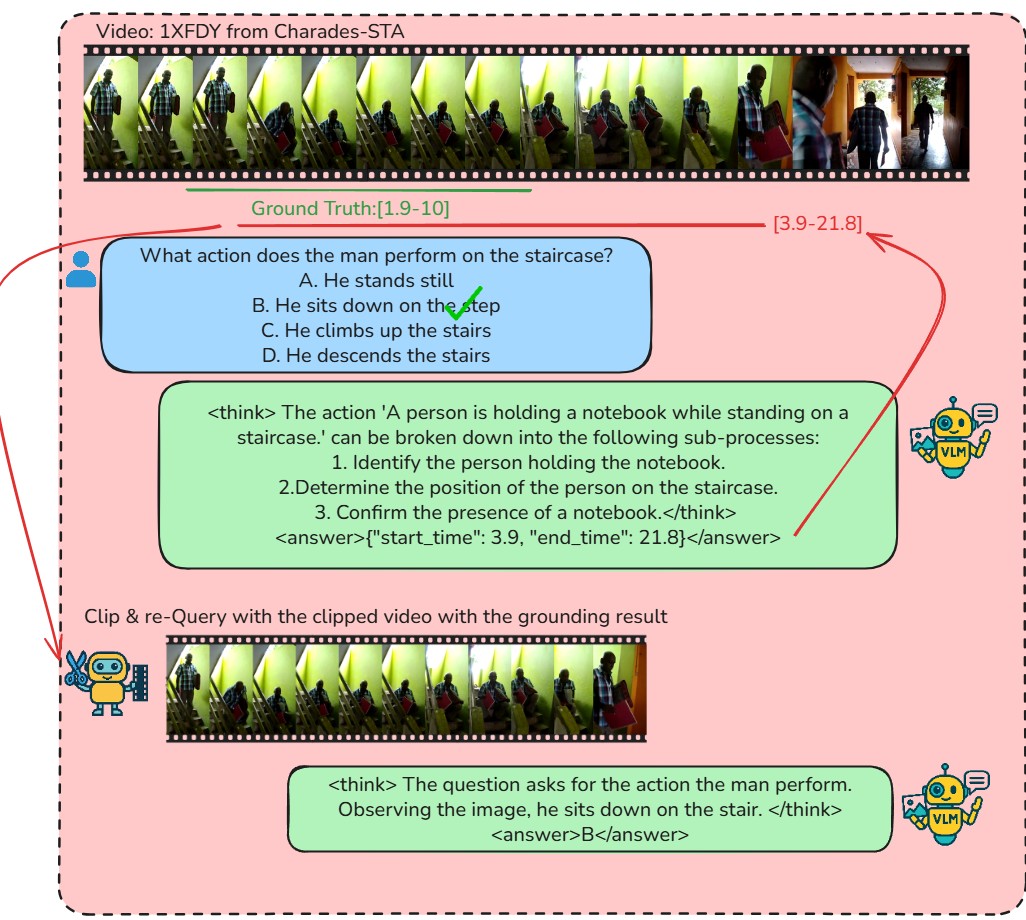

Figure 3: Qualitative demonstration of our two-stage reasoning and grounding pipeline on a sample video from Charade-STA.

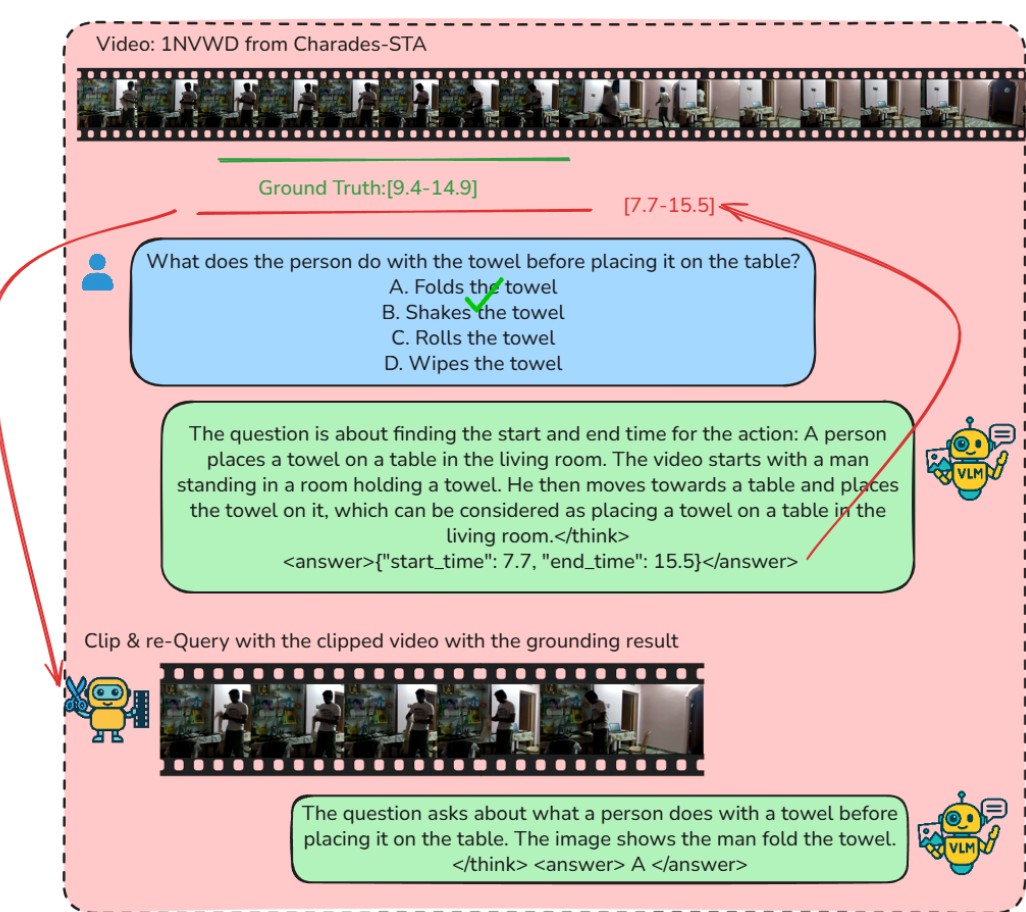

Figure 4: Qualitative demonstration of our two-stage reasoning and grounding pipeline on a sample video from Charade-STA.

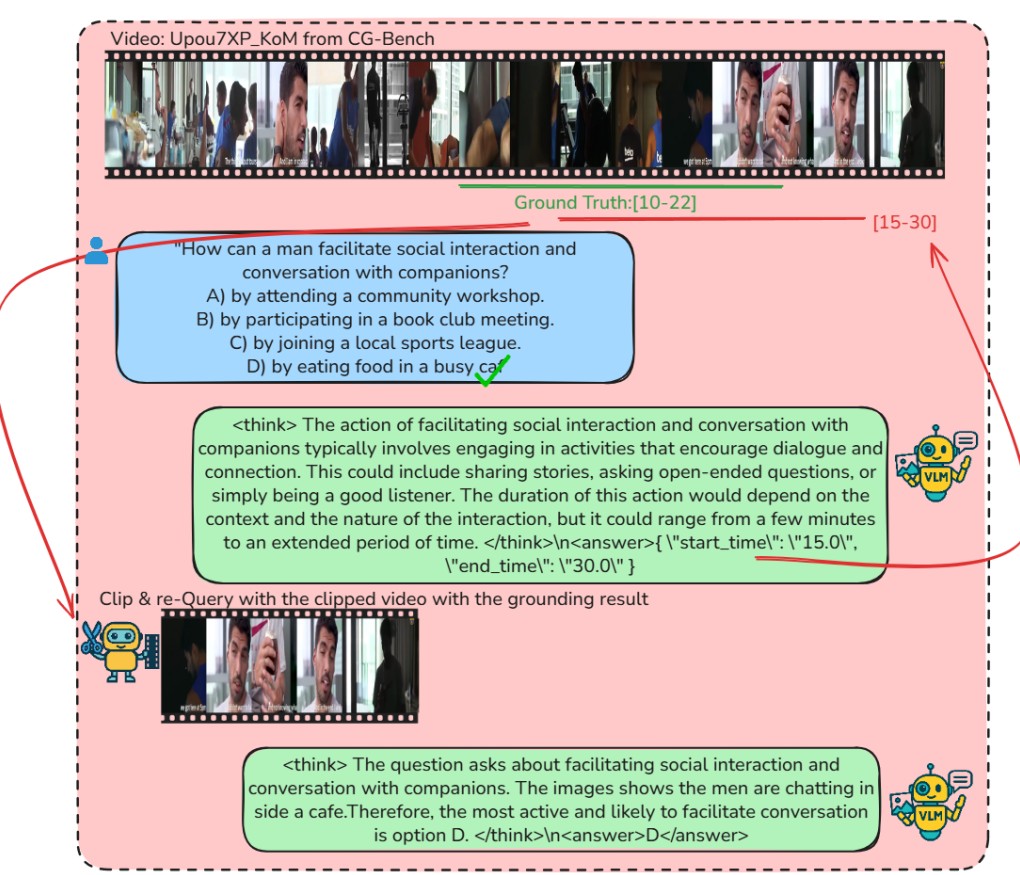

Figure 5: Qualitative demonstration of our two-stage reasoning and grounding pipeline on a sample video from CG-Bench.

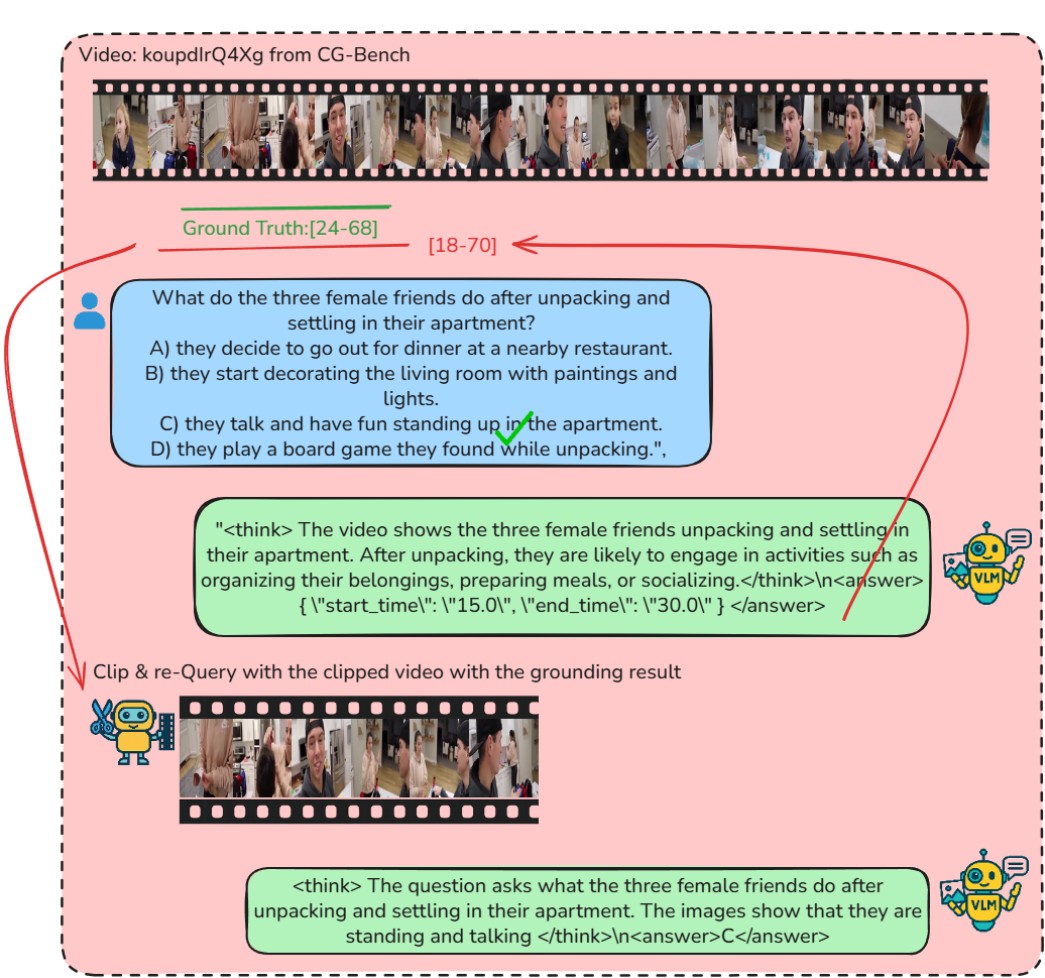

Figure 6: Qualitative demonstration of our two-stage reasoning and grounding pipeline on a sample video from CG-Bench.

