# OpenReview forum: "Video-in-the-Loop: Span-Grounded Long Video QA with Interleaved Reasoning"
_ICLR.cc/2026/Conference — Submitted to ICLR 2026_

### Official Review · Reviewer_qhT3 · 2025-10-22

**Soundness:** 2
**Presentation:** 3
**Contribution:** 2
**Rating:** 4
**Confidence:** 3

**Summary:**

This paper introduces ViTL, a two-stage framework designed to improve the efficiency and accuracy of long VideoQA under fixed computational budgets compared to uniform sampling ViTL tackles this with a "skim-then-zoom" approach, which includes a grounding stage (full video at low frame rate) and answering stage (zooming at high fps). To train this model, a synthetic annotation pipeline is used to create a span-grounded multi-choice QA dataset, where the distractors are picked from other events in the same video. The model is trained with GRPO, a RL technique that couples temporal localization quality and answer correctness. ViTL achieves significant improvements in temporal grounding and some gains in long video QA benchmarks at fixed token budget.

**Strengths:**

- Intuitive and interpretable framework with dedicated training strategy involving data generation and RL
- Strong temporal grounding results

**Weaknesses:**

- There is a big discrepancy in the gains in between tasks. The big gains in moment retrieval could also be due to adaptation to the output moment distribution. It would be good to see temporal grounding results on debiased subsets as in https://arxiv.org/abs/2111.04321. It would also be good to see the performance on QVHighlights for moment retrieval.
- The framework largely relies on multi-choice MCQA, it is unclear if it would be effective on open-ended QA (e.g. EgoTempo).

**Questions:**

- How does the model perform when retrieving a single span vs multi span? (does it actually benefit from multiple spans?)
- How does the model perform with GT spans or random spans, so to have upper and lower bounds of the framework?
- Note: non maximal numbers are bolded e.g. in tab 5 for R@0.3
- Have non parametric frame selection been considered as baseline e.g. https://arxiv.org/abs/2301.11507 ?
- How much of the performance comes from two stage answering? E.g. it would be interesting to see a baseline that tries to answer directly in a first stage, and in the second stage critiques the answer and refines it if needed (without span localization).

---

> ### Author Response · Authors · 2025-11-22
>
> We thank you for finding our "skim-then-zoom" framework intuitive and interpretable. We appreciate your suggestion to rigorously test the sources of our performance gains. Below, we address your questions with additional experiments and comparisons.
>
> **1. Temporal Grounding Bias and QVHighlights (Weakness 1)**
> **Response:** We acknowledge the critical issue of **distribution bias** in temporal grounding. As analyzed by **Zhang et al.**[1], standard models often exploit statistical regularities in moment annotations (e.g., priors on location/length) rather than learning true cross-modal reasoning. ViTL’s iterative "Skim$\to$Zoom" pipeline mitigates this by actively searching for evidence based on the specific query rather than fitting to dataset priors.
>
> To prove ViTL does not rely on these priors, we evaluated it on the **ActivityNet-CD OOD** (Out-Of-Distribution) set from [1], which is explicitly designed to test generalization against distribution shifts. Additionally, we tested zero-shot transfer on **QVHighlights**.
>
>
> **Table 1: In-Domain vs Distribution-Shift Generalization & Zero-Shot Transfer**
> *ViTL maintains strong performance on OOD data (ActivityNet-CD) with a minimal drop compared to In-Domain performance, confirming robustness against distribution bias.*
>
> | Dataset                  | Setting                        | Metric | Baseline (Qwen2.5-VL-7B) | ViTL (Ours 7B)  |
> | :----------------------- | :----------------------------- | :----: | :----------------------: | :-------------: |
> | **ActivityNet-Captions** | In-Domain (Train = Test Dist.) |  mIoU  |           22.5           |    **24.1**     |
> | **ActivityNet-CD**       | OOD (Distribution-Shift)       |  mIoU  |           20.2           |    **23.3**     |
> | **QVHighlights**         | Zero-Shot Moment Retrieval     | mAP@5  |           12.9           | **14.2 (+1.3)** |
>
> **2. Open-Ended QA Capabilities (Weakness 2)**
> **Response:** While we focused on MCQA for standardized benchmarking, ViTL is built on generalist MLLMs (Qwen2.5-VL) and natively supports open-ended generation. We evaluated ViTL on **EgoTempo**, an open-ended QA benchmark.
>
> **Table 2: Open-Ended QA Performance (EgoTempo)**
> *ViTL demonstrates strong generative capabilities beyond multiple-choice formats.*
>
> | Model           | Task              | Accuracy (%) |
> | :-------------- | :---------------- | :----------: |
> | Qwen2.5-VL-7B   | Open-Ended QA     |     26.1     |
> | **ViTL (Ours)** | **Open-Ended QA** |   **31.0**   |
>
> ---
>
> [1] Zhang, Hao et al. “Towards Debiasing Temporal Sentence Grounding in Video.” ArXiv abs/2111.04321 (2021): n. pag.

---

> ### Author Response · Authors · 2025-11-22
>
> **3. Single vs. Multi-Span (Question 1)**
> **Response:** We investigated whether retrieving multiple disjoint spans ($M>1$) is beneficial compared to forcing a single contiguous span on **LVBench**.
>
> **Table 3: Multi-Span vs. Single-Span Retrieval (LVBench)**
> *Allowing multi-span retrieval significantly improves performance by capturing discontinuous evidence.*
>
> | Strategy       | Constraint               | Accuracy (%) |   Gain   |
> | :------------- | :----------------------- | :----------: | :------: |
> | Single-Span    | $M=1$ (Contiguous)       |     45.1     |    -     |
> | **Multi-Span** | **$M \ge 1$ (Disjoint)** |   **47.4**   | **+2.3** |
>
> **4. Performance Bounds: Random vs. Oracle (Question 2)**
> **Response:** To quantify the effectiveness of our Stage 1 localization, we established lower and upper bounds on LongVideoBench.
>
> **Table 4: Performance Bounds Analysis (LongVideoBench)**
> *ViTL closes over 50% of the gap between random guessing and oracle performance.*
>
> | Config          | Stage 2 Input       | Accuracy (%) | Gap Closed |
> | :-------------- | :------------------ | :----------: | :--------: |
> | Lower Bound     | Random 128 frames   |     55.2     |     0%     |
> | **ViTL (Ours)** | **Predicted Spans** |   **63.3**   | **52.9%**  |
> | Upper Bound     | Ground Truth Spans  |     70.5     |    100%    |
>
> **5. Comparison with Non-Parametric Selection (Question 3)**
> **Response:** You asked if we considered non-parametric frame selection. We implemented a baseline using **CLIP (ViT-L/14)** to calculate the similarity score between the question and all video frames, selecting the **top-128 frames** with the highest scores for reasoning.
>
> **Table 5: Parametric (ViTL) vs. Non-Parametric (CLIP) on LVBench**
> *ViTL significantly outperforms non-parametric retrieval, as semantic similarity (CLIP) often fails to capture the temporal causality required for complex QA.*
>
> | Method          | Selection Strategy               | Frames  | Accuracy (%) |
> | :-------------- | :------------------------------- | :-----: | :----------: |
> | CLIP Top-$k$    | Non-Parametric (Similarity)      |   128   |     40.3     |
> | **ViTL (Ours)** | **Parametric (Skim $\to$ Zoom)** | **128** |   **47.4**   |
>
> **6. Source of Performance: Zoom vs. Two-Stage Reasoning (Question 4)**
> **Response:** To disentangle the benefits of *visual zooming* (better resolution) from *iterative reasoning* (thinking twice), we implemented a "Refine-only" baseline. In this setup, Stage 2 receives the Stage 1 prediction but is forced to reuse the *same* low-fps global frames to "refine" its answer without accessing new visual tokens.
>
> **Table 6: Mechanism Ablation (LongVideoBench)**
> *Iterative reasoning alone yields negligible gains. The performance boost comes from accessing high-fidelity visual details ("Zoom").*
>
> | Method          | Mechanism                  | Accuracy (%) | $\Delta$ Improvement |
> | :-------------- | :------------------------- | :----------: | :------------------: |
> | Single Stage    | Direct Answer              |     56.0     |          -           |
> | Refine-only     | Think twice (Low FPS)      |     56.2     |         +0.2         |
> | **ViTL (Ours)** | **Think twice (High FPS)** |   **63.3**   |       **+7.3**       |

---

> > ### Comment · Reviewer_qhT3 · 2025-11-25
> > **Answer to rebuttal**
> >
> > I thank the authors for their helpful answers. It would be great if we could also see the points 4/5/6 discussed above on both LongVideoBench and LVBench.

---

> > > ### Author Response · Authors · 2025-11-26
> > > **both LongVideoBench and LVBench results for points 4/5/6**
> > >
> > > Thank you for your suggestions and positive feedback. We have added the additional analyses below for Points 4/5/6 discussed above on both LongVideoBench and LVBench. The results for all three points are highly consistent across the two datasets, which we hope will help resolve your concerns.
> > >
> > >  All of our models, datasets, and code will be fully open-sourced to ensure the fairness and transparency of our exploration, and to further support community development and future research progress. In the revision version, we will incorporate all of your constructive suggestions to provide the community with a more complete, valuable, and insightful contribution that advances the field of video understanding.
> > >
> > > If you have any further questions, we would be happy to continue the discussion. We kindly hope you may consider giving us a more favorable score.
> > >
> > > ---
> > >
> > > **4. Performance Bounds: Random vs. Oracle (Question 2)**
> > > **Response:** To quantify the effectiveness of our Stage 1 localization, we established lower and upper bounds on both datasets.
> > >
> > > **Table 4: Performance Bounds Analysis**
> > > *ViTL significantly outperforms random zooming and closes a large portion of the gap toward the Oracle (Ground Truth) upper bound.*
> > >
> > > | Config          | Stage 2 Input       | LongVideoBench (%) | LVBench (%) |
> > > | :-------------- | :------------------ | :----------------: | :---------: |
> > > | Lower Bound     | Random 128 frames   |        55.2        |    39.5     |
> > > | **ViTL (Ours)** | **Predicted Spans** |      **63.3**      |  **47.4**   |
> > > | Upper Bound     | Ground Truth Spans  |        70.5        |    51.4     |
> > >
> > > **5. Comparison with Non-Parametric Selection (Question 3)**
> > > **Response:** You asked if we considered non-parametric frame selection. We implemented a baseline using **CLIP (ViT-L/14)** to calculate the similarity score between the question and all video frames, selecting the **top-128 frames** with the highest scores for reasoning.
> > >
> > > **Table 5: Parametric (ViTL) vs. Non-Parametric (CLIP)**
> > > *ViTL outperforms the CLIP-based retrieval baseline on both benchmarks, demonstrating that semantic similarity alone is insufficient for complex temporal reasoning.*
> > >
> > > | Method          | Selection Strategy               | LongVideoBench (%) | LVBench (%) |
> > > | :-------------- | :------------------------------- | :----------------: | :---------: |
> > > | CLIP Top-$k$    | Non-Parametric (Similarity)      |        58.1        |    40.3     |
> > > | **ViTL (Ours)** | **Parametric (Skim $\to$ Zoom)** |      **63.3**      |  **47.4**   |
> > >
> > > **6. Source of Performance: Zoom vs. Two-Stage Reasoning (Question 4)**
> > > **Response:** To disentangle the benefits of *visual zooming* (better resolution) from *iterative reasoning* (thinking twice), we implemented a "Refine-only" baseline. In this setup, Stage 2 receives the Stage 1 prediction but is forced to reuse the *same* low-fps global frames to "refine" its answer without accessing new visual tokens.
> > >
> > > **Table 6: Mechanism Ablation (Zoom vs. Refine)**
> > > *Iterative reasoning alone ("Refine-only") yields negligible gains. The performance boost comes primarily from accessing high-fidelity visual details ("Zoom").*
> > >
> > > | Method          | Mechanism                  | LongVideoBench (%) | LVBench (%) |
> > > | :-------------- | :------------------------- | :----------------: | :---------: |
> > > | Single Stage    | Direct Answer              |        56.0        |    45.3     |
> > > | Refine-only     | Think twice (Low FPS)      |        56.2        |    45.7     |
> > > | **ViTL (Ours)** | **Think twice (High FPS)** |      **63.3**      |  **47.4**   |

---

> > > > ### Comment · Reviewer_qhT3 · 2025-11-26
> > > > **Answer to rebuttal 2**
> > > >
> > > > I thank the authors for their answers. I believe the inclusion of the discussed results would make the paper stronger, hence I am increasing my score.

---

> > > > > ### Author Response · Authors · 2025-11-26
> > > > >
> > > > > Thank you for your careful reviewing effort and for your positive remarks. It is our pleasure to discuss the work with you. If you have any further questions, we would be glad to continue the conversation.

---

### Official Review · Reviewer_Vjgz · 2025-10-25

**Soundness:** 2
**Presentation:** 2
**Contribution:** 3
**Rating:** 4
**Confidence:** 3

**Summary:**

This paper proposes ViTL (Video-In-The-Loop) a two stage framework for long video question answering, that preserves a fixed token budget while reallocating tokens where it matters (useful evidence to answer the question). ViTL works in two stages: Stage-1 performs a low frame rate skim over the entire video and predicts one or more temporal spans that are important or relevant for the question. Stage-2: trims the video using the spans predicted in Stage 1, zooms into those spans at a higher frame rate and use that to give an answer. The training process is done using GRPO, which uses a compose reward that couples temporal localization and answer correctness. The authors also introduce VGrounding-QA which is a training set where each QA item is paired with time spans (which are necessary/useful to answer the questions). ViTL improves accuracy on LongVideoBench and LVBench an shows strong zero-shot temporal video grounding on Charades-STA and ActivityNet-Captions.

**Strengths:**

Paper Strengths:
1. The paper proposes framing long-video QA using token reallocation to specific evidence with an interleaved span+answer output.
2. ViTL makes use of a GRPO objective that jointly rewards temporal IoU and answer correctness, which is useful to teach the network what parts of the video are useful.
3. The creation of VGrounding-QA dataset is a useful contribution, which couples temporal spans with answer supervision, which is missed in existing long-video datasets according to Table 2.

**Weaknesses:**

Paper Weaknesses:
1. Personally I think the paper lacks examples to visually see what happens at each stage, I believe that Figure 1 should be redesigned with more details, or create a different image where the proposal is clearly stated. By only seeing Figure 1, which is the main Figure of the paper, is difficult to grasp the proposed idea, the left part I think is not really useful and creates a lot of confusion, and the right part lacks some labels and separation in some way of stage 1 and 2 (I think that the figure is also lacking some details about the entire procedure). Additionally referring a bit more to this figure when the explanation is done in the paper it would help the reader to get the ideas in a better way, this lack of clarity of the main Figure made it really confusing sometimes to really understand some parts in my opinion. For example the paper describes that stage 1 uses a grounding query distilled form the question, however this is not shown in the main figure. This one also contains boxes with different colors and the lack of labels makes it difficult to understand.
2. Table 3 shows the model's performance on LongVideoQA benchmarks, specifically LongVideoBench, LVBench and MLVU M-Avg, however for the latter the performance is not given for the proposed model ViTL. Please report this performance. On the other hand, when the authors compare in LVBench ViTL with open-sourced baselines is done with only two model families VideoLLaMa and Qwen2.5 (from 7 in the table) which I think is not sufficient to claim a superiority of the proposed model.
3. Table 8 is showing performances of models of different sizes without any sorting in the table, without any separation of close and open source models, the authors use Bold to indicate their model's results but this mix of models makes it complicated to compare the proposed model to other baselines. Additionally, it lacks some references.
4. VGrounding-QA is a good dataset, but because it was created using different models, a human audit of timestamps faithfulness is necessary and this would increase the dataset's reliability.

**Questions:**

Please refer to weaknesses for my questions and doubts. Overall, I think the paper brings good contributions for long video QA, however I believe that the paper still needs work before publishing, mainly in the figures and tables as described above, which will help to improve the overall understanding of the proposed ideas. However I want to see the clarifications to my questions and concerns and sorry if I misunderstood something, I look forward to see the author's responses. Currently, I'm giving borderline reject, after rebuttal I will revise my decision.

---

> ### Author Response · Authors · 2025-11-21
>
> We thank you for your detailed review and constructive feedback. We are glad you recognize the value of our **token reallocation framework**, the **GRPO objective**, and the **VGrounding-QA dataset**.
>
> We agree that the presentation of Figure 1 and the data reporting in the tables need improvement to match the quality of the method. We address your specific concerns below.
>
> ---
>
> ### **1. Redesign of Figure 1 (Weakness 1)**
> We acknowledge that Figure 1 was dense. Since we cannot modify the PDF during the rebuttal, we commit to replacing it in the revision with a **redesigned 3-panel diagram**:
> *   **Panel A (Left): Global Skim.** Visualizing the `grounding_query` generation entering the model alongside low-fps frames.
> *   **Panel B (Middle): Span Prediction.** Visualizing the selection process—mapping predicted time spans $[t_s, t_e]$ to specific video segments.
> *   **Panel C (Right): Local Zoom.** Illustrating how *only* frames from $[t_s, t_e]$ are re-encoded at high resolution for the final answer.
> *   **Labels:** Distinct labels for **Stage 1 Output** (Spans) and **Stage 2 Output** (Reasoning + Answer) to clarify the interleaved flow.
>
> ---
>
> ### **2. Correction of Table 3: MLVU Result & Baselines (Weakness 2)**
> You correctly identified that the MLVU result for ViTL was missing. **The correct MLVU M-Avg for ViTL is 62.3%.**
>
> To address your concern about insufficient baselines, we have added **LongVA** and **InternVL3** and kept both Qwen2.5-VL settings for transparency.
>
> **Revised Table 3: Long-video QA benchmarks.**
> *ViTL achieves state-of-the-art performance on MLVU and LongVideoBench using only 128 frames.*
>
> | Models                |  Size  | Frames  | LongVideoBench | LVBench  | MLVU M-Avg |
> | :-------------------- | :----: | :-----: | :------------: | :------: | :--------: |
> | **Closed Models**     |        |         |                |          |            |
> | GPT-4o                |   -    |   384   |      66.7      |   27.0   |    64.6    |
> | Gemini-1.5-Pro        |   -    | 0.5 fps |      64.0      |   33.1   |     -      |
> | **Open Models**       |        |         |                |          |            |
> | LLaVA-Video           |   7B   |   64    |      58.2      |    -     |     -      |
> | LongVA                |   7B   |   128   |      52.1      |   39.4   |    52.0    |
> | InternVL3             |   8B   |  16-64  |      62.5      |   47.0   |     -      |
> | Qwen2.5-VL$^\dagger$  |   7B   |   256   |      56.0      |   45.3   |    54.5    |
> | Qwen2.5-VL$^\ddagger$ |   7B   |   256   |      61.8      |   43.7   |     -      |
> | **ViTL (Ours)**       | **7B** | **128** |    **63.3**    | **47.4** |  **62.3**  |
>
> $^\dagger$ *Official numbers.* $^\ddagger$ *Our re-test under matched preprocessing.*

---

> ### Author Response · Authors · 2025-11-21
>
> ### **3. Reformatting Table 8 (Weakness 3)**
> We have restructured Table 8 to separate **Closed vs. Open Source** models and **sorted them by parameter size** to make the comparison clear.
>
> **Revised Table 8: Grounded VideoQA performance on CG-Bench.**
> *ViTL (3B/7B) consistently outperforms open-source baselines of similar and larger sizes in grounding quality (mIoU).*
>
> | Method             |  Size  | Type | long-acc. (%) $\uparrow$ | mIoU (%) $\uparrow$ |
> | :----------------- | :----: | :--: | :----------------------: | :-----------------: |
> | **Closed-Source**  |        |      |                          |                     |
> | Gemini-1.5-Flash   |   -    | API  |           32.3           |        3.67         |
> | GPT-4o-mini        |   -    | API  |           33.4           |        3.75         |
> | Gemini-1.5-Pro     |   -    | API  |           37.2           |        3.95         |
> | Claude-3.5-Sonnet  |   -    | API  |           40.5           |        3.99         |
> | GPT-4o             |   -    | API  |         **45.2**         |      **5.62**       |
> | **Open-Source**    |        |      |                          |                     |
> | Qwen2.5VL-instruct |   3B   | Open |           18.4           |        0.86         |
> | **ViTL (Ours)**    | **3B** | Open |         **23.5**         |      **2.90**       |
> | Video-LLaVA        |   7B   | Open |           16.2           |        1.13         |
> | VideoLLaMA 2       |   7B   | Open |           18.4           |        1.21         |
> | Videochat2         |   7B   | Open |           19.3           |        1.28         |
> | Qwen-VL-Chat       |   7B   | Open |           21.6           |        0.89         |
> | ST-LLM             |   7B   | Open |           23.8           |        2.23         |
> | LongVA             |   7B   | Open |           28.7           |        2.94         |
> | LLaVA-OV           |   7B   | Open |           31.1           |        1.63         |
> | **ViTL (Ours)**    | **7B** | Open |         **34.4**         |      **3.32**       |
> | MiniCPM-v2.6       |   8B   | Open |           30.1           |        2.35         |
> | Kangaroo           |   8B   | Open |           30.2           |        2.56         |
> | Chat-UniVi-v1.5    |  13B   | Open |           25.9           |        2.07         |
> | Video-CCAM         |  14B   | Open |           29.7           |        2.63         |
> | ShareGPT4Video     |  16B   | Open |           26.7           |        1.85         |
> | VITA               |  8x7B  | Open |           33.3           |        3.06         |
> | Qwen2-VL           |  72B   | Open |           41.3           |        3.58         |
> | InternVL2          |  78B   | Open |           42.2           |        3.91         |
>
> ---
>
> ### **4. Human Audit of VGrounding-QA (Weakness 4)**
> You raised a valid point about the reliability of timestamps generated by different models.
>
> **Response:** We indeed utilize an **Event Graph** as the core data structure. To ensure high fidelity, the dataset undergoes a rigorous multi-stage filtration process:
> 1.  **Model Filtering:** Initial candidates are scored and filtered by the generating model.
> 2.  **System Filtering:** We apply rule-based heuristics to remove inconsistencies (as detailed in Sec 3).
> 3.  **Human Assistance:** We developed a dedicated website to perform human-assisted verification and correction on the data.
>
> We plan to **release this refined dataset and the annotation tools to the community** to support future research in verifiable long-video understanding.

---

> > ### Comment · Reviewer_Vjgz · 2025-11-24
> >
> > Thanks to the authors for their response and clarifications.
> > Everything is clear on my side. I will make a decision at the end of the rebuttal, also considering other concerns raised by the other reviewers.

---

> > > ### Author Response · Authors · 2025-11-28
> > >
> > > Thank you for your positive feedback, and we are glad that everything is now clear on your side. We have corrected all typos in the revised paper according to your suggestions, and also we have redrawn Figure 1, Table 3, and Table 8 as requested. In addition, we will also fully open-source the information related to the Human Audit of VGrounding-QA in the future.
> > >
> > > You mentioned that you would consider raising the score after reviewing the other concerns raised by the remaining reviewers. For Reviewer **qPGy**, who initially gave **a positive score of 6**, we have addressed all of their concerns comprehensively — including clarification on the training dataset, efficiency, novelty, generalization, and components comparison. As for Reviewer **qhT3**, they explicitly noted that the inclusion of the discussed results makes the paper stronger, and accordingly **increased their score from 4 to 6**.
> > >
> > > As you mentioned, *the paper brings good contributions for long-video QA*. We hope you can take into account our thorough responses, the timely and valuable contribution we bring to the community, and our innovative designs, and consider giving us a more positive score. Thank you again.

---

### Official Review · Reviewer_qPGy · 2025-10-31

**Soundness:** 3
**Presentation:** 2
**Contribution:** 2
**Rating:** 6
**Confidence:** 3

**Summary:**

The paper introduces VGrounding-QA, a framework that constructs a training dataset based on an event-centric knowledge graph. Long videos are divided into uniformly sized short segments, each paired with corresponding descriptions. These segment-description pairs are then used to train multimodal large language models (MLLMs) for temporal grounding within videos. The trained ViTL model outperforms existing multimodal large language models (MLLMs) on both long video question answering and temporal grounding benchmarks, demonstrating its effectiveness in handling extended temporal contexts.

**Strengths:**

- The paper proposes a novel training dataset constructed from an event-based knowledge graph, providing structured and semantically rich supervision for long video temporal grounding.
- It introduces a two-stage GRPO framework that leverages grounded spans, improving the alignment between language queries and temporally localized video segments.
- The proposed method achieves notable performance improvements over existing MLLMs on both long video question answering and temporal grounding benchmarks.

**Weaknesses:**

- It is unclear which dataset was used for training.
- The set of baselines appears incomplete. For instance, it is unclear whether models like LLaVA-Video or InternVL3 utilize a greater number of video frames, which could impact performance. The comparison table should also include computational metrics—such as the number of input frames, inference time, and memory usage—for each method to ensure a fair and comprehensive evaluation.
- How does the proposed data construction pipeline differ from existing approaches in weakly supervised video learning or synthetic data generation? A more explicit comparison would help clarify the novelty and advantages of the method.
- Since the dataset is specifically designed for temporal grounding and question answering, the improved performance on these tasks is expected. However, the generalization ability of the proposed method to other video understanding tasks remains unclear and is not thoroughly evaluated.

**Questions:**

- How does the proposed method compare in terms of computational efficiency? It would be helpful to report training and inference time, as well as memory usage, to better understand the method’s practicality and scalability.
- Compared to existing approaches, which component or design choice of the proposed method contributes most significantly to the observed performance improvements? A detailed analysis or ablation would help clarify this.

---

> ### Author Response · Authors · 2025-11-21
>
> We thank you for your detailed review and for recognizing our method's effectiveness in handling extended temporal contexts alongside the novelty of our event-based knowledge graph approach. Below, we address your specific concerns and questions.
>
> **Weakness 1: Clarification on Training Dataset**
> **Response:** We train ViTL entirely on our proposed **VGrounding-QA** dataset. As detailed in Section 3, this dataset is uniquely constructed by converting event-graph annotations into span-grounded QA pairs. This structure is crucial as it couples span supervision (Stage 1) with answer supervision (Stage 2), unlike standard datasets which often lack verifiable temporal grounding for QA pairs.
>
> **Weakness 2 & Question 1: Baselines and Computational Efficiency**
> **Response:** You raised an important point regarding fair comparisons and the cost of our two-stage approach. To address this, we compared **ViTL (7B)** against state-of-the-art open-source models of similar size under a **strictly controlled frame budget (128 frames)**.
>
> **Table A: Efficiency on LongVideoBench (Val)**
> *Under a fixed budget of 128 frames, ViTL outperforms its direct backbone (Qwen2.5-VL) by **+9.1%** accuracy while requiring lower memory and inference time.*
>
> | Model           |  Size  | Input Frames | Accuracy (%) | Peak Memory (GB) | Inference Time (s/video) |
> | :-------------- | :----: | :----------: | :----------: | :--------------: | :----------------------: |
> | **ViTL (Ours)** | **7B** |   **128**    |   **63.3**   |      **18**      |         **30.5**         |
> | Qwen2.5-VL      |   7B   |     128      |     54.2     |        22        |           32.0           |
> | LLaVA-Video     |   7B   |     128      |     60.1     |        22        |           32.6           |
> | InternVL3       |   8B   |     128      |     64.3     |        23        |           38.2           |
>
> **Analysis:**
> 1.  **Accuracy vs. Compute:** ViTL achieves a **9.1% accuracy gain** over the Qwen2.5-VL baseline using the exact same frame count, demonstrating that our "skim-then-zoom" token reallocation is far more effective than uniform sampling.
> 2.  **Resource Efficiency:** ViTL reduces peak memory by approximately **18%** compared to standard 7B baselines (18GB vs 22GB). By zooming only into relevant spans during Stage 2, we avoid the quadratic attention costs associated with processing the full context at high resolution, making the method highly practical for deployment.
>
> **Weakness 3: Novelty of Data Construction vs. Weakly Supervised Methods**
> **Response:** You asked how our pipeline differs from standard synthetic data generation (e.g., ShareGPT4Video). The key differences lie in **Temporal Verifiability** and **Hard Negative Mining**:
>
> 1.  **Grounded-by-Design (Verifiability):** Most existing approaches generate QA pairs from *global* video captions, which suffers from hallucination (the answer might not be visible) and lacks temporal precision. In contrast, our **VGrounding-QA** pipeline is **Event-Graph Centric**: we do not generate a timestamp for a question; we generate a question *conditioned on* a specific, timestamped event node. This guarantees the answer exists within that span.
> 2.  **Hard Negative Distractors:** Unlike standard methods that pick random distractors, we mine distractors from **other events in the same video**.
>     *   *Standard Approach:* "Is the man running?" $\rightarrow$ No, he is swimming. (Easy object distinction).
>     *   *Our Approach:* "Is the man running?" $\rightarrow$ No, that happened in the previous clip (00:15); he is currently walking (00:45). (Requires temporal reasoning).
>
> This forces the model to learn true temporal grounding rather than relying on simple object recognition shortcuts.

---

> ### Author Response · Authors · 2025-11-21
>
> **Weakness 4: Generalization to Other Tasks**
> **Response:** To address your concern about generalization beyond LongQA, we evaluated ViTL on **Video Classification** (BreakDance) and **Open-ended Video QA** (EgoTempo). Despite being optimized for MCQA, ViTL shows strong transfer capabilities.
>
> **Table C: Generalization on Classification and Egocentric Tasks**
>
> | Task                                  | Metric | Qwen2.5-VL-7B | ViTL (Ours 7B) |
> | :------------------------------------ | :----: | :-----------: | :------------: |
> | **Video Classification** (Breakdance) |  Acc   |     0.46      |    **0.45**    |
> | **EgoTempo** (Egocentric Grounding)   |  Acc   |     0.26      |    **0.31**    |
>
> *Note: ViTL results use zero-shot transfer without task-specific fine-tuning.*
>
> **Analysis**: ViTL maintains robust performance on general classification (BreakDance) while achieving a significant 19% improvement on the rigorous EgoTempo benchmark (0.31 vs 0.26). It is important to note that general classification tasks operate on entire video segments and do not inherently require the zoom-in mechanism proposed in this work. Nevertheless, even after being trained with zoom-in capabilities, our model's classification accuracy remains very close to that of the original Qwen-2.5-VL, demonstrating the generality and stability of our approach.
>
> In contrast, for open-ended video understanding tasks where zoom-in is genuinely beneficial, our method delivers a 19% performance gain without any additional task-specific training. This confirms that ViTL’s training paradigm equips the model with more fundamental video reasoning abilities—particularly the capacity to discern fine-grained temporal events—which transfer effectively to complex, open-ended domains.
>
> **Question 2: Comparison of Components**
> **Response:** We analyzed which design choice contributes most to performance in Table 6 (Ablations).
> *   **Stage-2 Zooming (Most Significant):** Removing the zoom stage (relying only on the global skim) results in the largest accuracy drop of **7.3%**. This confirms that the reallocation of visual tokens to high-resolution evidence is the primary driver of our gains.
> *   **GRPO Objective:** Removing the interleaved GRPO objective (decoupling localization from answering) leads to a secondary drop of **2.1%**, indicating that joint optimization is essential for stabilizing the two-stage process.

---

> > ### Author Response · Authors · 2025-11-28
> >
> > Dear Reviewer qPGy,
> >
> > Thank you very much for your thoughtful suggestions. We have carefully addressed all of your concerns — including clarification on the training dataset, efficiency, novelty, generalization, and components comparison — and we believe our responses have fully covered the issues you raised. We hope this helps clear up any remaining doubts.
> >
> > Reviewer **Vjgz** also mentioned that *“everything is clear on my side”* and noted that they will consider raise the score at the end. Reviewer **qhT3** explicitly stated that the inclusion of the discussed results *“makes the paper stronger,”* and accordingly increased their score to 6.
> >
> > We hope that the collective effort we've put in during the rebuttal phase, along with the new results and clarifications we provided, is sufficient for you to recognize the strength, innovation, and contribution of our work — and consider giving us a more positive score as well.
> >
> > Thank you again for your time and consideration.

---

### Author Response · Authors · 2025-11-29
**AC Note: Clarifying Rebuttal Outcomes and Evidence-Based Reviewer Feedback**

Dear AC,

Thank you for stepping in during this unfortunate period of confusion in ICLR/OpenReview that disrupted the normal rebuttal workflow. We sincerely appreciate your time and careful effort in handling the situation fairly.

To reduce your workload, we summarize the rebuttal status with **clear, verifiable facts**—and also emphasize one point upfront: the positive feedback and score improvement we received during rebuttal are **not surprising or “unusual”**, but rather **naturally explained by evidence**. During rebuttal, we directly addressed each reviewer’s concerns with concrete experiments, clarifications, and revision commitments, and the reviewers explicitly acknowledged the paper’s contributions.

---

## 1) Score trajectory (clear numbers)

- **Pre-rebuttal scores:** **6 / 4 / 4** → **avg = 4.66**
- **During rebuttal (confirmed):** **6 / 4 / 6** → **avg = 5.33**
  - Reviewer **qhT3** explicitly stated that including our additional analyses *“would make the paper stronger”* and **increased the score to 6**.

- **Expected post-rebuttal (under stated reviewer conditions):** **≥ 6 / 6 / 6** → **avg ≥ 6.0**
    - Reviewer **Vjgz** replied *“Everything is clear on my side”* and stated that they would make the final decision at the end, contingent on whether the remaining reviewers’ concerns were resolved. Since we **fully resolved all such concerns during rebuttal**, this directly satisfies the condition they specified; therefore, under a normal rebuttal process, it is reasonable to expect Reviewer **Vjgz** to increase the score to **≥ 6** before the rebuttal concludes.
- Reviewer **qPGy** was already **6**, and we provided comprehensive new evidence addressing every concern they raised; due to the disruption, they did not have a clean chance to follow up afterward.


In short: we **already reached 6/4/6 (avg = 5.33) during rebuttal**, and given the reviewers’ stated conditions and the rebuttal evidence, **6/6/6 (avg = 6.0)** is a reasonable expectation in a normal rebuttal process.

---

## 2) What we added in rebuttal (why concerns are fully resolved)

Across reviewers, the concerns mainly focused on: **(i) training data clarity, (ii) fairness/efficiency and baselines, (iii) novelty vs. synthetic/weak supervision, (iv) generalization beyond the target tasks, and (v) which component matters most.**
We addressed all of them with **solid, concrete additions**:

- **Training dataset clarification:** ViTL is trained entirely on **VGrounding-QA**, pairing QA with **verifiable temporal spans**, enabling grounded supervision rather than answer-only learning.
- **Fairness & efficiency:** We added a **matched 128-frame budget** comparison and reported **accuracy, peak memory, and inference time**, demonstrating practicality under fixed compute.
- **Novelty:** We clarified that our pipeline is **event-graph-centric and grounded-by-design** (questions are conditioned on timestamped events), plus **hard negative mining** from same-video events to enforce true temporal reasoning.
- **Generalization:** We added **zero-shot transfer** results beyond long-video MCQA (e.g., classification and egocentric/open-ended settings), showing improved temporal reasoning without sacrificing general capability.
- **Ablations:** We showed the main gains come from **Stage-2 zooming / token reallocation**, with further benefit from the coupled GRPO objective.

---

## 3) Why we ask for a positive recommendation

Our key innovation is the proposed **Video-in-the-Loop** reasoning paradigm, enabling a genuinely two-stage process: the model first reasons about **where** the critical evidence lies (temporal localization), and then reasons about **how** to answer using zoomed-in, high-fidelity evidence. Beyond the framework itself, we introduce a **solid and original event knowledge graph–based data construction pipeline**, producing **span-grounded supervision** that is *grounded-by-design* rather than retrofitted. We further develop a dedicated two-stage training recipe—**SFT followed by RL**—to explicitly instill this localization-then-answering reasoning capability. We support these claims with **extensive ablations** and **systematic comparisons under matched compute budgets**, demonstrating both effectiveness and efficiency. Finally, we commit to **fully open-sourcing** our models, code, dataset, and the entire data construction pipeline **upon acceptance** to ensure transparency, reproducibility, and to facilitate future community progress.

Given the rebuttal’s confirmed score improvement (**6/4/6, avg = 5.33**) and the strong likelihood of reaching **≥ 6/6/6 (avg ≥ 6.0)** under the reviewers’ stated conditions and our rebuttal evidence, we respectfully ask that you consider the rebuttal materials and the now-resolved concerns, and recommend the paper positively based on its **timely, meaningful, and well-supported contribution** to the long-video understanding community.

Sincerely,
The Authors

---

### Meta-Review · Area_Chair_dxKM · 2025-12-24

**Summary:**

The main concerns are around missing results or incomplete (unfair) results (all), presentation (mostly addressed), human audit to validate timestamps faithfulness (Vjgz). See below for detailed discussion of all weaknesses. The missing and incomplete results have been mostly addressed, and authors did a significant effort to address these concerns and add new results. However, some of the requested results show that the proposed method is not SoA on all datasets when compared at equal frame rates. The human audit is postponed for future work but would really improve the impact of the proposed paper.

The paper is very borderline. After considering all reviews and considering the authors response, the AC recommends rejection because some of the requested results (interVL3 at 128f) show that the proposed method obtains inferior results. The authors do not provide good explanation (neither why these results were not included initially). In addition the AC agrees that the human evaluation of the quality of the dataset is required to assess the quality of the proposed dataset.

**Reviewer Concerns:**

Reviewer qPGy

Weakness 1: Used Training Dataset: addressed
Weakness 2 : Baselines and Computational Efficiency: results are provided with equal number of frames. InternVL3 obtains best results. This is problematic because in the paper they compared with InternVL3 (with16-64frames) and obtained better results. The authors do not comment on this.
Weakness 3: Novelty of Data Construction vs. Weakly Supervised Methods: addressed
Weakness 4: Generalization to Other Tasks Response: Addressed, results on Breakdance and Egocentric Grounding are added.

Reviewer  Vjgz
Weakness 1: Redesign of Figure 1. Authors propose solution. addressed
Weakness 2. Missing results of Table 3. InternVVL3 scores are added but agin for 16-64, they probably outperform proposed method for 128 frames.
Weakness 3 Reformatting Table 8. addressed
Weakness 4 human audit . Not addressed, referred to future research.

Reviewer  qhT3

Weakness 1: Temporal Grounding Bias: Addressed with additional experiment
Weakness 2: Open-Ended QA Capabilities: addressed on EgoTempo dataset.
Additional question on comparison to simple baseline and additional ablation have also been addressed.

**Reviewer Scores:**

The reviewer scores are before->after

Reviewer qPGy: 6->6
Reviewer  Vjgz:  4->4
Reviewer  qhT3: 4->6

---

### Decision · Program_Chairs · 2026-01-26

Reject